# Conditional Simulation Using Diffusion Schrödinger Bridges

**Yuyang Shi**[1]      **Valentin De Bortoli**[2]      **George Deligiannidis**[1]      **Arnaud Doucet**[1]

[1]Department of Statistics, University of Oxford, UK
[2]ENS, PSL University, Paris, France

## Abstract

Denoising diffusion models have recently emerged as a powerful class of generative models. They provide state-of-the-art results, not only for unconditional simulation, but also when used to solve conditional simulation problems arising in a wide range of inverse problems. A limitation of these models is that they are computationally intensive at generation time as they require simulating a diffusion process over a long time horizon. When performing unconditional simulation, a Schrödinger bridge formulation of generative modeling leads to a theoretically grounded algorithm shortening generation time which is complementary to other proposed acceleration techniques. We extend the Schrödinger bridge framework to conditional simulation. We demonstrate this novel methodology on various applications including image super-resolution, optimal filtering for state-space models and the refinement of pre-trained networks. Our code can be found at `https://github.com/vdeborto/cdsb`.

## 1 INTRODUCTION

*Score-Based Generative Models* (SGMs), also known as denoising diffusion models, are a class of generative models that have become recently very popular as they provide state-of-the-art performance; see *e.g.* Chen et al. [2021a], Ho et al. [2020], Song et al. [2021b], Saharia et al. [2021], Dhariwal and Nichol [2021]. Existing SGMs proceed as follows. First, noise is gradually added to the data using a time-discretized diffusion so as to provide a sequence of perturbed data distributions eventually approximating an easy-to-sample reference distribution, typically a multivariate Gaussian. Second, one approximates the corresponding time-reversed denoising diffusion using neural network ap-

proximations of the logarithmic derivatives of the perturbed data distributions known as scores; these approximations are obtained using denoising score matching techniques [Vincent, 2011, Hyvärinen, 2005]. Finally, the generative model is obtained by initializing this reverse-time process using samples from the reference distribution [Ho et al., 2020, Song et al., 2021b].

In many applications, one is not interested in unconditional simulation but the generative model is used as an implicit prior $p_{\text{data}}(x)$ on some parameter $X$ (e.g. image) in a Bayesian inference problem with a likelihood function $g(y^{\text{obs}}|x)$ for observation $Y = y^{\text{obs}}$. SGMs have been extended to address such tasks, see *e.g.* Song et al. [2021b], Saharia et al. [2021], Batzolis et al. [2021], Tashiro et al. [2021]. In this conditional simulation case, one only requires being able to simulate from the joint distribution of data and synthetic observations $(X, Y) \sim p_{\text{data}}(x)g(y|x)$. As in the unconditional case, the time-reversal of the noising diffusion is approximated using neural network estimates of its scores, the key difference being that this network admits not only $x$ but also $y$ as an input. Sampling from the posterior $p(x|y^{\text{obs}}) \propto p_{\text{data}}(x)g(y^{\text{obs}}|x)$ is achieved by simulating the time-reversal using the scores evaluated at $Y = y^{\text{obs}}$.

However, performing unconditional or conditional simulation using SGMs is computationally expensive as, to obtain a good approximation of the time-reversed diffusion, one needs to run the forward noising diffusion long enough to converge to the reference distribution. Many techniques have been proposed to accelerate simulation including *e.g.* knowledge distillation [Luhman and Luhman, 2021, Salimans and Ho, 2022], non-Markovian forward process and subsampling [Song et al., 2021a], optimized noising diffusions and improved numerical solvers [Jolicoeur-Martineau et al., 2021, Dockhorn et al., 2022, Kingma et al., 2021, Watson et al., 2022]. In the unconditional scenario, reformulating generative modeling as a Schrödinger bridge (SB) problem provides a principled theoretical framework to accelerate simulation time complementary to most other acceleration techniques [De Bortoli et al., 2021]. The SB solution is the

*Accepted for the 38th Conference on Uncertainty in Artificial Intelligence* (UAI 2022).

finite time process which is the closest in terms of Kullback–Leibler (KL) discrepancy to the forward noising process used by SGMs but admits as marginals the data distribution at time $t = 0$ and the reference distribution at time $t = T$. The time-reversal of the SB thus enables unconditional generation from the data distribution. However, the use of the SB formulation has not yet been developed in the context of conditional simulation.

The contributions of this paper are as follows.

- We develop conditional SB (CSB), an original SB formulation for conditional simulation.
- By adapting the Diffusion SB algorithm of De Bortoli et al. [2021] to our setting, we propose an iterative algorithm, Conditional Diffusion SB (CDSB), to approximate the solution to the CSB problem.
- CDSB performance is demonstrated on various examples. In particular, we propose the first application of score-based techniques to optimal filtering in state-space models.

## 2 SCORE-BASED GENERATIVE MODELING

### 2.1 UNCONDITIONAL SIMULATION

Assume we are given samples from some data distribution with positive density[1] $p_{\text{data}}$ on $\mathbb{R}^d$. Our aim is to provide a generative model to sample new data from $p_{\text{data}}$. SGMs achieve this as follows. We gradually add noise to data samples, i.e. we consider a Markov chain $x_{0:N} = \{x_k\}_{k=0}^N \in \mathcal{X} = (\mathbb{R}^d)^{N+1}$ of joint density

$$p(x_{0:N}) = p_0(x_0) \prod_{k=0}^{N-1} p_{k+1|k}(x_{k+1}|x_k), \quad (1)$$

where $p_0 = p_{\text{data}}$ and $p_{k+1|k}$ are Markov transition densities inducing the following marginal densities $p_{k+1}(x_{k+1}) = \int p_{k+1|k}(x_{k+1}|x_k)p_k(x_k)\mathrm{d}x_k$. These transition densities are selected such that $p_N(x_N) \approx p_{\text{ref}}(x_N)$ for large $N$, where $p_{\text{ref}}$ is an easy-to-sample *reference* density. In practice we set $p_{\text{ref}}(x_N) = \mathcal{N}(x_N; 0, \text{Id})$, while $p_{k+1|k}(x_{k+1}|x_k) = \mathcal{N}(x_{k+1}; x_k - \gamma_{k+1}x_k; 2\gamma_{k+1} \text{Id})$ for $\gamma_k > 0, \gamma_k \ll 1$ so $x_{0:N}$ is a time-discretized Ornstein–Uhlenbeck diffusion (see supplementary for details).

The main idea behind SGMs is to obtain samples from $p_0$ by exploiting the backward decomposition of (1)

$$p(x_{0:N}) = p_N(x_N) \prod_{k=0}^{N-1} p_{k|k+1}(x_k|x_{k+1}),$$

i.e. by sampling $X_N \sim p_N(x_N)$ then sampling $X_k \sim p_{k|k+1}(x_k|X_{k+1})$ for $k \in \{N-1, \ldots, 0\}$, we obtain $X_0 \sim p_0(x_0)$. In practice, we know neither $p_N$ nor the

[1]We assume here that all distributions admit a positive density w.r.t. Lebesgue measure.

backward transition densities $p_{k|k+1}$ for $k \in \{0, ..., N-1\}$ and therefore this ancestral sampling procedure cannot be implemented exactly. We thus approximate $p_N$ by $p_{\text{ref}}$ and $p_{k|k+1}$ using a Taylor expansion approximation

$$p_{k|k+1}(x_k|x_{k+1}) \approx \mathcal{N}(x_k; B_{k+1}(x_{k+1}), 2\gamma_{k+1} \text{Id}),$$

where $B_{k+1}(x) = x + \gamma_{k+1}\{x + 2\nabla \log p_{k+1}(x)\}$. Finally, we approximate the score terms $\nabla \log p_k$ using denoising score matching methods [Hyvärinen, 2005, Vincent, 2011, Song et al., 2021b]. Since $p_k(x_k) = \int p_0(x_0)p_{k|0}(x_k|x_0)\mathrm{d}x_0$, it follows that $\nabla \log p_k(x_k) = \mathbb{E}[\nabla_{x_k} \log p_{k|0}(x_k|X_0)]$, where the expectation is w.r.t. to the distribution of $X_0$ given $x_k$. We learn a neural network approximation $\mathbf{s}_{\theta^\star}(k, x_k) \approx \nabla \log p_k(x_k)$ by minimizing w.r.t. $\theta$ the loss

$$\mathbb{E}[\sum_{k=1}^N \lambda_k ||\mathbf{s}_\theta(k, X_k) - \nabla_{x_k} \log p_{k|0}(X_k|X_0)||^2],$$

where $\lambda_k > 0$ is a weighting coefficient [Ho et al., 2020, Song et al., 2021b] and the expectation is w.r.t. $p(x_{0:N})$. Once we have estimated $\theta^\star$ from noisy data, we start by first sampling $X_N \sim p_{\text{ref}}(x_N)$ and then sampling $X_k \sim \hat{p}_{k|k+1}(x_k|X_{k+1})$ for $\hat{p}_{k|k+1}$ as in $p_{k|k+1}$ but with $\nabla \log p_{k+1}(X_{k+1})$ replaced by $\mathbf{s}_{\theta^\star}(k+1, X_{k+1})$. Under regularity assumptions, the resulting $X_0$ can be shown to be approximately distributed according to $p_0 = p_{\text{data}}$ if $p_N \approx p_{\text{ref}}$ [De Bortoli et al., 2021, Theorem 1].

### 2.2 CONDITIONAL SIMULATION

We now consider the scenario where we have samples from $p_0 = p_{\text{data}}$ and are interested in generating samples from the posterior $p(x|y^{\text{obs}}) \propto p_0(x)g(y^{\text{obs}}|x)$ for some observation $Y = y^{\text{obs}} \in \mathcal{Y}$. Here it is assumed that it is possible to sample synthetic observations from $Y|(X = x) \sim g(y|x)$ but the expression of $g(y|x)$ might not be available.

In this case, conditional SGMs (CSGMs) proceed as follows; see e.g. Saharia et al. [2021], Batzolis et al. [2021], Li et al. [2022], Tashiro et al. [2021]. For any realization $Y = y$, we consider a Markov chain of the form (1) but initialized using $X_0 \sim p(x|y)$ instead of $p_0(x)$. Obviously it is not possible to simulate this chain but this will not prove necessary. This chain induces for $k \geq 0$ the marginals denoted $p_{k+1}(x_{k+1}|y)$ which satisfy $p_{k+1}(x_{k+1}|y) = \int p_{k+1|k}(x_{k+1}|x_k)p_k(x_k|y)\mathrm{d}x_k$ for $p_0(x_0|y) = p(x_0|y)$. Similarly to the unconditional case, to perform approximate ancestral sampling from this Markov chain, we need to sample from $p_{k|k+1}(x_k|x_{k+1}, y) \approx \mathcal{N}(x_k; B_{k+1}(x_{k+1}, y), 2\gamma_{k+1} \text{Id})$ where $B_{k+1}(x, y) = x + \gamma_{k+1}\{x + 2\nabla \log p_{k+1}(x|y)\}$. We can again estimate these score terms using

$$\nabla \log p_k(x_k|y) = \mathbb{E}[\nabla_{x_k} \log p_{k|0}(x_k|X_0)],$$

where the expectation is w.r.t. to the distribution of $X_0$ given $(X_k, Y) = (x_k, y)$. In this case, we learn again a neural

network approximation $\mathbf{s}_{\theta^\star}(k, x_k, y) \approx \nabla \log p_k(x_k|y)$ by minimizing w.r.t. $\theta$ the loss

$$\mathbb{E}[\textstyle\sum_{k=1}^N \lambda_k ||\mathbf{s}_\theta(k, X_k, Y) - \nabla_{x_k} \log p_{k|0}(X_k|X_0)||^2],$$

where the expectation is w.r.t. $p(x_{0:N})g(y|x_0)$ which we can sample from. Once the neural network is trained, we simulate from the posterior $p(x|y^{\text{obs}}) \propto p_0(x)g(y^{\text{obs}}|x)$ for any observation $Y = y^{\text{obs}}$ as follows: sample first $X_N \sim p_{\text{ref}}(x_N)$ and then $X_k \sim \hat{p}_{k|k+1}(x_k|X_{k+1}, y^{\text{obs}})$ where this density is similar to $p_{k|k+1}(x_k|X_{k+1}, y^{\text{obs}})$ but with $\nabla \log p_{k+1}(X_{k+1}|y^{\text{obs}})$ replaced by $\mathbf{s}_{\theta^\star}(k + 1, X_{k+1}, y^{\text{obs}})$. The resulting sample $X_0$ will be approximately distributed according to $p(x|y^{\text{obs}})$. This scheme can be seen as an amortized variational inference procedure.

# 3 SCHRÖDINGER BRIDGES AND GENERATIVE MODELING

For SGMs to work well, we must diffuse the process long enough so that $p_N \approx p_{\text{ref}}$. The SB methodology introduced in [De Bortoli et al., 2021] allows us to mitigate this problem. We refer to Chen et al. [2021b] for recent reviews on the SB problem. We first recall how the SB problem can be applied to perform unconditional simulation.

Consider the *forward* density $p(x_{0:N})$ given by (1), describing the process adding noise to the data. We want to find the joint density $\pi^\star(x_{0:N})$ such that

$$\pi^\star = \arg\min_\pi \{\text{KL}(\pi|p) \; : \; \pi_0 = p_{\text{data}}, \; \pi_N = p_{\text{ref}}\}, \quad (2)$$

where $\pi_0$, resp. $\pi_N$, is the marginal of $X_0$, resp. $X_N$, under $\pi$. A visualization of the SB problem (2) is provided in Figure 1a. Were $\pi^\star$ available, we would obtain a generative model by ancestral sampling: sample $X_N \sim p_{\text{ref}}(x_N)$, then $X_k \sim \pi^\star_{k|k+1}(x_k|X_{k+1})$ for $k \in \{N-1, \ldots, 0\}$.

The SB problem does not admit a closed-form solution but it can be solved numerically using Iterative Proportional Fitting (IPF) [Kullback, 1968]. This algorithm defines the following recursion initialized at $\pi^0 = p$ given in (1):

$$\pi^{2n+1} = \arg\min_\pi \{\text{KL}(\pi|\pi^{2n}) \; : \; \pi_N = p_{\text{ref}}\},$$
$$\pi^{2n+2} = \arg\min_\pi \{\text{KL}(\pi|\pi^{2n+1}) \; : \; \pi_0 = p_{\text{data}}\}.$$

De Bortoli et al. [2021], Vargas et al. [2021] showed that the IPF iterates admit a representation suited to numerical approximation. Indeed, if we denote $p^n = \pi^{2n}$ and $q^n = \pi^{2n+1}$, then $p^0(x_{0:N}) = p(x_{0:N})$ and

$$q^n(x_{0:N}) = p_{\text{ref}}(x_N) \textstyle\prod_{k=0}^{N-1} q^n_{k|k+1}(x_k|x_{k+1}),$$
$$p^{n+1}(x_{0:N}) = p_{\text{data}}(x_0) \textstyle\prod_{k=0}^{N-1} p^{n+1}_{k+1|k}(x_{k+1}|x_k),$$

where $q^n_{k|k+1} = p^n_{k|k+1}$ and $p^{n+1}_{k+1|k} = q^n_{k+1|k}$. To summarize, at step $n = 0$, $q^0$ is the backward process obtained

by reversing the dynamics of $p^0$ initialized at time $N$ from $p_{\text{ref}}$. The forward process $p^1$ is then obtained from the reversed dynamics of $q^0$ initialized at time 0 from $p_{\text{data}}$, and so on. Note that $q^0$ corresponds to the unconditional SGM described in Section 2.1.

## 3.1 DIFFUSION SCHRÖDINGER BRIDGE

Similarly to SGMs, one can approximate the time-reversals appearing in the IPF iterates using score matching ideas. If $p^n_{k+1|k}(x'|x) = \mathcal{N}(x'; x + \gamma_{k+1}f^n_k(x), 2\gamma_{k+1}\,\text{Id})$, with $f^0_k(x) = -x$, we approximate the reverse-time transitions by $q^n_{k|k+1}(x|x') \approx \mathcal{N}(x; x' + \gamma_{k+1}b^n_{k+1}(x'), 2\gamma_{k+1}\,\text{Id})$, where $b^n_{k+1}(x') = -f^n_k(x') + 2\nabla \log p^n_{k+1}(x')$; and next $p^{n+1}_{k+1|k}(x'|x) \approx \mathcal{N}(x'; x + \gamma_{k+1}f^{n+1}_k(x), 2\gamma_{k+1}\,\text{Id})$, where $f^{n+1}_k(x) = -b^n_{k+1}(x) + 2\nabla \log q^n_k(x)$. The drifts $b^n_{k+1}, f^{n+1}_k$ could be estimated by approximating $\{\nabla \log p^i_{k+1}(x)\}_{i=0}^n$, $\{\nabla \log q^i_k(x)\}_{i=0}^n$ using score matching. However this is too expensive both in terms of compute and memory. De Bortoli et al. [2021] instead directly approximate the mean of the Gaussians using neural networks, $\mathbf{B}_\theta$ and $\mathbf{F}_\phi$, by generalizing the score matching approach, i.e. $q^n_{k|k+1}(x|x') = \mathcal{N}(x; \mathbf{B}_{\theta^n}(k + 1, x'), 2\gamma_{k+1}\,\text{Id})$ and $p^n_{k+1|k}(x'|x) = \mathcal{N}(x'; \mathbf{F}_{\phi^n}(k, x), 2\gamma_{k+1}\,\text{Id})$, where $\theta^n$ is obtained by minimizing

$$\ell^b_n(\theta) = \mathbb{E}_{p^n}[\textstyle\sum_k ||\mathbf{B}_\theta(k + 1, X_{k+1}) - G_{n,k}(X_k, X_{k+1})||^2],$$

for $G_{n,k}(x, x') = x' + \mathbf{F}_{\phi^n}(k, x) - \mathbf{F}_{\phi^n}(k, x')$, and $\phi^{n+1}$ by minimizing

$$\ell^f_{n+1}(\phi) = \mathbb{E}_{q^n}[\textstyle\sum_k ||\mathbf{F}_\phi(k, X_k) - H_{n,k}(X_k, X_{k+1})||^2],$$

for $H_{n,k}(x, x') = x + \mathbf{B}_{\theta^n}(k+1, x') - \mathbf{B}_{\theta^n}(k+1, x)$. This implementation of IPF, referred to as Diffusion SB (DSB), is presented in the supplementary; see Vargas et al. [2021], Chen et al. [2022] for alternative numerical schemes. After we have learned $\theta^L$ using $L$ DSB iterations, we sample $X_N \sim p_{\text{ref}}(x_N)$ and then set $X_k = \mathbf{B}_{\theta^L}(k + 1, X_{k+1}) + \sqrt{2\gamma_{k+1}}Z_{k+1}$ with $Z_k \overset{\text{i.i.d.}}{\sim} \mathcal{N}(0, \text{Id})$ to obtain $X_0$ approximately distributed from $p_{\text{data}}$.

## 3.2 LINK WITH OPTIMAL TRANSPORT

It can be shown that the solution $\pi^\star$ of the SB problem (2), $\pi^\star(x_{0:N}) = \pi^{s,\star}(x_0, x_N)p_{|0,N}(x_{1:N-1}|x_0, x_N)$ where $\pi^{s,\star}(x_0, x_N)$ is the marginal of $\pi^\star(x_{0:N})$ at times 0 and $N$. In this case, (2) reduces to the static SB problem

$$\pi^{s,\star} = \arg\min_{\pi^s} \{\text{KL}(\pi^s|p_{0,N}) \; : \; \pi^s_0 = p_{\text{data}}, \; \pi^s_N = p_{\text{ref}}\}.$$

The static SB problem can be interpreted as an entropy-regularized optimal transport problem between $p_{\text{data}}$ and $p_{\text{ref}}$, with regularized transportation cost $\mathbb{E}_{\pi^s}[-\log p_{N|0}(X_N|X_0)] - H(\pi^s)$. When $p_{N|0}(x_N|x_0) =$

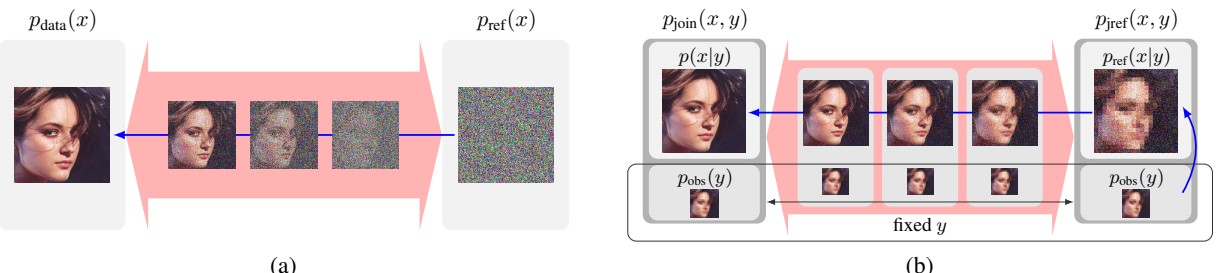

Figure 1: (a) An unconditional Schrödinger bridge (SB) between $p_{\text{data}}(x)$ and $p_{\text{ref}}(x)$; (b) our proposed conditional Schrödinger bridge (CSB) on the extended space between $p_{\text{join}}(x, y)$ and $p_{\text{jref}}(x, y)$. The blue arrows denote the direction of the generative procedure at simulation time.

$\mathcal{N}(x_N; x_0, \sigma^2)$ as in Song and Ermon [2019], the transportation cost $-\log p_{N|0}(x_N|x_0)$ reduces to the quadratic cost $\frac{1}{2\sigma^2}\|x_0 - x_n\|^2$ up to a constant. In other words, the static SB solution $\pi^{s,\star}$ not only transports samples $X_N \sim p_{\text{ref}}$ into samples from the data distribution $p_{\text{data}}$, but also seeks to minimize an entropy-regularized Wasserstein distance of order 2. The regularization strength is controlled by the variance $\sigma^2$. Similar properties hold for the time-discretized Ornstein–Uhlenbeck diffusion defined by (1) in Section 2.1.

# 4 CONDITIONAL DIFFUSION SCHRÖDINGER BRIDGE

We now want to use SBs for conditional simulation, i.e. to be able sample from a posterior distribution $p(x|y^{\text{obs}}) \propto p_{\text{data}}(x)g(y^{\text{obs}}|x)$ assuming only that it is possible to sample $(X, Y) \sim p_{\text{data}}(x)g(y|x)$. In this case, an obvious approach would be to consider the SB problem where we replace $p_{\text{data}}(x)$ by the posterior $p(x|y^{\text{obs}})$, i.e.

$$\pi^\star = \arg\min_\pi \left\{ \text{KL}(\pi|p_{y^{\text{obs}}}) : \pi_0 = p(\cdot|y^{\text{obs}}), \ \pi_N = p_{\text{ref}} \right\}, \tag{3}$$

where $p_{y^{\text{obs}}}(x_{0:n}) := p(x_0|y^{\text{obs}}) \prod_{k=0}^{N-1} p_{k+1|k}(x_{k+1}|x_k)$ is the forward noising process. However, DSB is not applicable here as it requires sampling from $p(x_0|y^{\text{obs}})$ at step 0.

We propose instead to solve an amortized problem. Let us introduce $p_{\text{join}}(x, y) = p_{\text{data}}(x)g(y|x) = p(x|y)p_{\text{obs}}(y)$ and $p_{\text{jref}}(x, y) = p_{\text{ref}}(x)p_{\text{obs}}(y)$ where $p_{\text{obs}}(y) = \int p_{\text{data}}(x)g(y|x)\mathrm{d}x$. We are interested in finding the transition kernel $\pi^{c,\star} = (\pi_y^{c,\star})_{y \in \mathcal{Y}}$, where $\pi_y^{c,\star}$ defines a distribution on $\mathcal{X} = (\mathbb{R}^d)^{N+1}$ for each $y \in \mathcal{Y}$, satisfying

$$\pi^{c,\star} = \arg\min_{\pi^c} \{ \mathbb{E}_{Y \sim p_{\text{obs}}}[\text{KL}(\pi_Y^c || p_Y)] :$$
$$\pi_0^c \otimes p_{\text{obs}} = p_{\text{join}}, \ \pi_N^c \otimes p_{\text{obs}} = p_{\text{jref}} \}. \tag{4}$$

This corresponds to an averaged version of (3) over the distribution $p_{\text{obs}}(y)$ of $Y$. The first constraint $\pi_{y,0}^{c,\star}(x_0)p_{\text{obs}}(y) = p_{\text{join}}(x_0, y) = p(x_0|y)p_{\text{obs}}(y)$ ensures that $\pi_{y,0}^{c,\star}(x_0) = p(x_0|y)$, $p_{\text{obs}}$-almost surely. Similarly $\pi_{y,N}^{c,\star}(x_N) =$

$p_{\text{ref}}(x_N)$, $p_{\text{obs}}$-almost surely. Hence, to obtain a sample from $p(x|y^{\text{obs}})$ for a given $Y = y^{\text{obs}}$, we can sample $X_N \sim p_{\text{ref}}(x_N)$ then $X_k|X_{k+1} \sim \pi_{y^{\text{obs}},k|k+1}^{c,\star}(x_k|X_{k+1})$ for $k = N-1, ..., 0$ and $X_0$ is a sample from $p(x|y^{\text{obs}})$.

We show here that (4) can be reformulated as a SB on an extended space, which we will refer to as Conditional SB (CSB), so the theoretical results for existence and uniqueness of the solution to the SB problem apply.

**Proposition 1.** *Consider the following SB problem*

$$\bar{\pi}^\star = \arg\min_{\bar{\pi}} \{ \text{KL}(\bar{\pi}|\bar{p}) : s.t. \ \bar{\pi}_0 = p_{\text{join}}, \ \bar{\pi}_N = p_{\text{jref}} \}, \tag{5}$$

*where we define* $\bar{p}(x_{0:N}, y_{0:N}) := p_{y_0}(x_{0:N})\bar{p}_{\text{obs}}(y_{0:N})$ *with* $\bar{p}_{\text{obs}}(y_{0:N}) := p_{\text{obs}}(y_0) \prod_{k=0}^{N-1} \delta_{y_k}(y_{k+1})$ *and* $p_{y_0}$ *is the forward process defined below* (3). *If* $\text{KL}(\bar{\pi}^\star|\bar{p}) < +\infty$ *then* $\bar{\pi}^\star = \pi^{c,\star} \otimes \bar{p}_{\text{obs}}$ *where* $\pi^{c,\star}$ *solves* (4).

We provide an illustration of the CSB problem (5) in Figure 1b. Under $\bar{p}$, the $Y$-component is sampled at time 0 according to $p_{\text{obs}}$ and then is kept constant until time $N$ while the $X$-component is initialized at $p(x|y_0)$ and then diffuses according to $p_{k+1|k}(x_{k+1}|x_k)$.

Contrary to (3), we can adapt DSB to solve numerically the CSB problem (5) as both the distributions $p_{\text{join}}$ and $p_{\text{jref}}$ can be sampled. The resulting algorithm is called Conditional DSB (CDSB). It approximates the following IPF recursion

$$\bar{\pi}^{2n+1} = \arg\min_{\bar{\pi}} \{ \text{KL}(\bar{\pi}|\bar{\pi}^{2n}) : \bar{\pi}_N = p_{\text{jref}} \},$$
$$\bar{\pi}^{2n+2} = \arg\min_{\bar{\pi}} \{ \text{KL}(\bar{\pi}|\bar{\pi}^{2n+1}) : \bar{\pi}_0 = p_{\text{join}} \}$$

initialized at $\bar{\pi}^0 = \bar{p}$. For $\bar{p}^n = \bar{\pi}^{2n}$ and $\bar{q}^n = \bar{\pi}^{2n+1}$, we have the following representation of the IPF iterates.

**Proposition 2.** *Assume that* $\text{KL}(p_{\text{join}} \otimes p_{\text{jref}}|\bar{p}_{0,N}) < +\infty$. *Then we have* $\bar{p}^0(x_{0:N}, y_{0:N}) = \bar{p}(x_{0:N}, y_{0:N})$ *and for any* $n > 0$, $\bar{q}^n(x_{0:N}, y_{0:N}) = \bar{p}_{\text{obs}}(y_{0:N})\bar{q}^n(x_{0:N}|y_N)$, $\bar{p}^{n+1}(x_{0:N}, y_{0:N}) = \bar{p}_{\text{obs}}(y_{0:N})\bar{p}^{n+1}(x_{0:N}|y_0)$ *with*

$$\bar{q}^n(x_{0:N}|y_N) = p_{\text{ref}}(x_N) \prod_{k=0}^{N-1} \bar{p}_{k|k+1}^n(x_k|x_{k+1}, y_N),$$
$$\bar{p}^{n+1}(x_{0:N}|y_0) = p(x_0|y_0) \prod_{k=0}^{N-1} \bar{q}_{k+1|k}^n(x_{k+1}|x_k, y_0).$$

Here we simplify notation and write $Y$ for all the random variables $Y_0, Y_1, ..., Y_N$ as they are all equal almost surely under $\bar{p}^n$ and $\bar{q}^n$. We approximate the transition kernels as in DSB and refer to the supplementary for more details. In particular, the transition kernels satisfy $\bar{q}_{k|k+1}^n(x|x', y) = \mathcal{N}(x; \mathbf{B}_{\theta^n}^y(k+1, x'), 2\gamma_{k+1} \mathrm{Id})$ and $\bar{p}_{k+1|k}^n(x'|x, y) = \mathcal{N}(x'; \mathbf{F}_{\phi^n}^y(k, x), 2\gamma_{k+1} \mathrm{Id})$, where $\theta^n$ is obtained by minimizing

$$\ell_n^b(\theta) = \mathbb{E}_{\bar{p}^n}[\sum_k \|\mathbf{B}_\theta^Y(k+1, X_{k+1}) - G_{n,k}^Y(X_k, X_{k+1})\|^2] \tag{6}$$

for $G_{n,k}^y(x, x') = x' + \mathbf{F}_{\phi^n}^y(k, x) - \mathbf{F}_{\phi^n}^y(k, x')$ and $\phi^{n+1}$ by minimizing

$$\ell_{n+1}^f(\phi) = \mathbb{E}_{\bar{q}^n}[\sum_k \|\mathbf{F}_\phi^Y(k, X_k) - H_{n,k}^Y(X_k, X_{k+1})\|^2], \tag{7}$$

$$H_{n,k}^y(x, x') = x + \mathbf{B}_{\theta^n}^y(k+1, x') - \mathbf{B}_{\theta^n}^y(k+1, x).$$

The resulting CDSB scheme is summarized in Algorithm 1 where $Z_k^j, \tilde{Z}_k^j \overset{\text{i.i.d.}}{\sim} \mathcal{N}(0, \mathrm{Id})$. After $L$ iterations of CDSB, we have learned $\theta^L$. For any observation $Y = y^{\text{obs}}$, we can then sample $X_N \sim p_{\text{ref}}(x_N)$ and then compute $X_k = \mathbf{B}_{\theta^L}^{y^{\text{obs}}}(k+1, X_{k+1}) + \sqrt{2\gamma_{k+1}} Z_{k+1}$ with $Z_k \overset{\text{i.i.d.}}{\sim} \mathcal{N}(0, \mathrm{Id})$ for $k = N-1, ..., 0$. The resulting sample $X_0$ will be approximately distributed from $p(x|y^{\text{obs}})$.

---

**Algorithm 1** Conditional Diffusion Schrödinger Bridge

1: **for** $n \in \{0, \dots, L\}$ **do**
2:     **while** not converged **do**
3:         Sample $\{X_k^j\}_{k,j=0}^{N,M}, \{Y^j\}_{j=0}^M$ where
        $X_0^j \sim p_{\text{data}}, Y^j \sim g(\cdot|X_0^j)$, and
        $X_{k+1}^j = \mathbf{F}_{\phi^n}^{Y^j}(k, X_k^j) + \sqrt{2\gamma_{k+1}} Z_{k+1}^j$
4:         Compute $\hat{\ell}_n^b(\theta^n)$ approximating (6)
5:         $\theta^n \leftarrow$ Gradient Step$(\hat{\ell}_n^b(\theta^n))$
6:     **end while**
7:     **while** not converged **do**
8:         Sample $\{X_k^j\}_{k,j=0}^{N,M}, \{Y^j\}_{j=0}^M$ where
        $X_N^j \sim p_{\text{ref}}, Y^j \sim p_{\text{obs}}$, and
        $X_k^j = \mathbf{B}_{\theta^n}^{Y^j}(k+1, X_{k+1}^j) + \sqrt{2\gamma_{k+1}} \tilde{Z}_{k+1}^j$
9:         Compute $\hat{\ell}_{n+1}^f(\phi^{n+1})$ approximating (7)
10:       $\phi^{n+1} \leftarrow$ Gradient Step$(\hat{\ell}_{n+1}^f(\phi^{n+1}))$
11:     **end while**
12: **end for**
13: **Output:** $(\theta^L, \phi^{L+1})$

---

# 5 CDSB IMPROVEMENTS

## 5.1 CONDITIONAL REFERENCE MEASURE

In standard SGMs and for the unconditional SB, we typically select $p_{\text{ref}}(x) = \mathcal{N}(x; 0, \sigma_{\text{ref}}^2 \mathrm{Id})$. However, initializing ancestral sampling from random noise to eventually obtain

samples from $p(x|y)$ can be inefficient as $y$ already contains useful information about $X$. Fortunately, it is easy to use a joint reference measure of the form $p_{\text{jref}}(x, y) = p_{\text{ref}}(x|y)p_{\text{obs}}(y)$ instead of $p_{\text{jref}}(x, y) = p_{\text{ref}}(x)p_{\text{obs}}(y)$ in CSB and CDSB. The only modification in Algorithm 1 is that line 8 becomes $Y^j \sim p_{\text{obs}}(y), X_N^j \sim p_{\text{ref}}(x|Y^j)$. In some interesting scenarios, we can select $p_{\text{ref}}(x|y)$ as an approximation to $p(x|y)$ in order to accelerate the sampling process. This means we construct a CSB between $p(x|y)$ and its approximation $p_{\text{ref}}(x|y)$, instead of between $p(x|y)$ and noise. We refer to this extension of CDSB as CDSB-C.

As a simple example, consider obtaining super-resolution (SR) image samples from a low-resolution image $Y = y$. Assume that $y$ has been suitably upsampled to have the same dimensionality as $X$. In this case, $y$ itself can serve as an approximate initialization for sampling $X_N$. A simple model is to take $p_{\text{ref}}(x|y) = \mathcal{N}(x; y, \sigma_{\text{ref}}^2 \mathrm{Id})$ with $\sigma_{\text{ref}}^2 = \rho \sigma_{x|y}^2$, where $\rho$ is a variance inflation parameter and $\sigma_{x|y}^2$ is an estimate of the conditional variance of $X$ given $Y$. See Figure 1b for an illustration. In our experiments, we also explore other $p_{\text{ref}}(x|y)$ obtained using the Ensemble Kalman Filter (EnKF) as well as neural network models.

## 5.2 CONDITIONAL FORWARD PROCESS

To accelerate the convergence of IPF, we also have the flexibility to make the initial forward noising process dynamics dependent on $Y = y$, i.e. $p_y(x_{0:N}) = p(x_0|y) \prod_{k=0}^{N-1} p_{k+1|k}(x_{k+1}|x_k, y)$. As shown below, it is beneficial to initialize $p_y$ close to the CSB solution $\pi_y^{c,\star}$.

**Proposition 3.** *For any $n \in \mathbb{N}$ with $n \geq 1$, we have*

$$\mathbb{E}[\mathrm{KL}(\pi_{Y,0}^{c,n}|p(\cdot|Y))] \leq \frac{2}{n}\mathbb{E}[\mathrm{KL}(\pi_Y^{c,\star}|p_Y)],$$

*where for any $n \in \mathbb{N}$, $\bar{\pi}^n = \bar{p}_{\text{obs}} \otimes \pi^{c,n}$ is the $n^{\text{th}}$ IPF iterate and the expectations are w.r.t. $Y \sim p_{\text{obs}}$.*

As a result, we should choose the initial forward noising process $p_y$ such that its terminal marginal $p_{y,N}$ targets $p_{\text{ref}}(\cdot|y)$. However, contrary to diffusion models, we recall that our framework does not strictly require $p_{y,N} \approx p_{\text{ref}}(\cdot|y)$ to provide approximate samples from the posterior of interest.

For tractable $p_{\text{ref}}(x|y)$, we can define $p_y(x_{0:N})$ using an unadjusted Langevin dynamics; i.e. $p_{k+1|k}(x'|x, y) = \mathcal{N}(x'; x + \gamma_{k+1} \nabla \log p_{\text{ref}}(x|y), 2\gamma_{k+1} \mathrm{Id})$. In the case $p_{\text{ref}}(x|y) = \mathcal{N}(x; \mu(y), \sigma^2(y) \mathrm{Id})$, this reduces to a discretized Ornstein–Uhlenbeck process admitting $p_{\text{ref}}(x|y)$ as limiting distribution as $\gamma \to 0$ and $N \to \infty$ [Durmus and Moulines, 2017].

## 5.3 FORWARD-BACKWARD SAMPLING

When we use an unconditional $p_{\text{ref}}(x)$, our proposed method also shares connections with the conditional transport

methodology developed by Marzouk et al. [2016], Spantini et al. [2022]. They propose methods to learn a deterministic invertible transport map $\mathcal{S}(x,y): \mathcal{X} \times \mathcal{Y} \to \mathcal{X}$ which maps samples from $p(x|y)$ to $p_{\text{ref}}(x)$. To sample from $p(x|y^{\text{obs}})$, one samples $X^{\text{ref}} \sim p_{\text{ref}}(x)$, then transports back the sample through the inverse map $X^{\text{pos}} = \mathcal{S}(\cdot, y^{\text{obs}})^{-1}(X^{\text{ref}})$.

As noted by Spantini et al. [2022], an alternative method to sample from $p(x|y^{\text{obs}})$ consists of first sampling $(X,Y) \sim p_{\text{join}}$, then following the two-step transformation $\hat{X}^{\text{ref}} = \mathcal{S}(X,Y)$, $\hat{X}^{\text{pos}} = \mathcal{S}(\cdot, y^{\text{obs}})^{-1}(\hat{X}^{\text{ref}})$. By definition of $\mathcal{S}$, $\hat{X}^{\text{ref}}$ is also distributed according to $p_{\text{ref}}$. However, since the transport map $\mathcal{S}$ may be imperfect in practice, this sampling strategy provides the advantage of cancellation of errors between $\mathcal{S}$ and $\mathcal{S}(\cdot, y^{\text{obs}})^{-1}$.

We also explore an analogous forward-backward sampling scheme in our framework, which first samples $(X,Y) \sim p_{\text{join}}$, followed by sampling $\hat{X}_N \sim \bar{p}^L_{N|0}(x_N|X,Y)$ through the forward half-bridge, then $\hat{X}_0 \sim \bar{q}^L_{0|N}(x_0|\hat{X}_N, y^{\text{obs}})$ through the backward half-bridge. Since $\bar{q}^L$ is the approximate time-reversal of $\bar{p}^L$, this strategy shares similar advantages as the method of Spantini et al. [2022] when the half-bridge $\bar{q}^L(x_{0:N}|y^{\text{obs}})$ does not solve the CSB problem exactly. We call this extension CDSB-FB.

# 6 RELATED WORK

**Approximate Bayesian computation (ABC)**, also known as likelihood-free inference, has been developed to approximate the posterior when the likelihood is intractable but one can simulate synthetic data from it; see *e.g.* [Beaumont, 2019]. However, these methods typically require knowing the prior, while CDSB only needs to have access to joint samples and learns about the posterior directly. For tasks such as image inpainting, the prior is indeed implicit.

**Schrödinger bridges** techniques to perform both static and sequential Bayesian inference for state-space models have been developed by Bernton et al. [2019] and Reich [2019]. However, these methods require being able to evaluate pointwise an unnormalized version of the target posterior distribution contrary to the CDSB-based methods developed here.

**Conditional transport**. Performing conditional simulation by learning a transport map between joint distributions on $X, Y$ having the same $Y$-marginals (as $p_{\text{join}}$ and $p_{\text{ref}}$) has been first proposed by Marzouk et al. [2016]. Various techniques have been subsequently developed to approximate such maps such as polynomial or radial basis representations [Marzouk et al., 2016, Baptista et al., 2020], Generative Adversarial Networks [Kovachki et al., 2021, Zhou et al., 2022] or normalizing flows [Kruse et al., 2021]. CDSB also fits into this framework, but instead utilizes stochastic transport maps. Recently, Taghvaei and Hosseini [2022] have also proposed independently using conditional transport ideas to perform optimal filtering for state-space models.

**Conditional SGMs**. SGMs have been applied to perform posterior simulation, primarily for images, as described in Section 2.2 and references therein. An alternative line of work for image editing [Song and Ermon, 2019, Choi et al., 2021, Chung et al., 2021, Meng et al., 2022] utilizes the denoising property of SGMs to iteratively denoise noisy versions of a reference image $y$ while restricted to retain particular features of $y$. However, $p_{\text{ref}}(x) = \mathcal{N}(x; 0, \sigma^2_{\text{ref}} \text{Id})$ so image generation is started from noise and typically hundreds or thousands of refinement steps are required. Our framework can incorporate in a principled way information given by $y$ in the reverse process's initialization (see Section 5.1). Recently Zheng et al. [2022], Lu et al. [2022] have also proposed suitable choices for $p_{\text{ref}}(x)$ or $p_{\text{ref}}(x|y)$ to shorten the diffusion process. In comparison, the CDSB framework is more flexible and allows for general $p_{\text{ref}}(x|y)$ which can be non-Gaussian and different from the initial forward diffusion's terminal distribution $p_N(x_N|y)$. For instance, we explore using noiseless pre-trained super-resolution models as $p_{\text{ref}}(x|y)$ in Section 7.3.2, where CDSB further improves the SR samples closer to the data distribution. Finally, for linear Gaussian inverse problems, Kadkhodaie and Simoncelli [2021], Kawar et al. [2021, 2022] develop efficient methodologies using unconditional SGMs when the linear degradation model and the Gaussian noise level are known.

**SGM acceleration techniques**. Many techniques have been proposed to accelerate SGMs and CSGMs. For example, Luhman and Luhman [2021], Salimans and Ho [2022] propose to learn a distillation network on top of SGM models, while Song et al. [2021a] perform a subsampling of the timesteps in a variational setting. Watson et al. [2022] optimize the timesteps with a fixed budget using dynamic programming. Xiao et al. [2021] perform multi-steps denoising using GANs while Dockhorn et al. [2022] consider underdamped Langevin dynamics as forward process. We emphasize that many of these techniques are complementary to and can be readily applied in the SB setting; *e.g.* one could distill the last CDSB network $\mathbf{B}^y_{\theta^L}$. Additionally, SB and CSB provide a framework to perform few-step sampling.

# 7 EXPERIMENTS

## 7.1 2D SYNTHETIC EXAMPLES

We first demonstrate the validity and accuracy of our method using the two-dimensional examples of Kovachki et al. [2021]. We consider three nonlinear, non-Gaussian examples for $p_{\text{join}}(x,y)$: define $p_{\text{obs}}(y) = \text{Unif}(y; [-3,3])$ for all examples and $p(x|y)$ is defined through

Example 1: $X = \tanh(Y) + Z$, $\quad Z \sim \Gamma(1, 0.3)$,
Example 2: $X = \tanh(Y + Z)$, $\quad Z \sim \mathcal{N}(0, 0.05)$,
Example 3: $X = Z \tanh(Y)$, $\quad Z \sim \Gamma(1, 0.3)$.

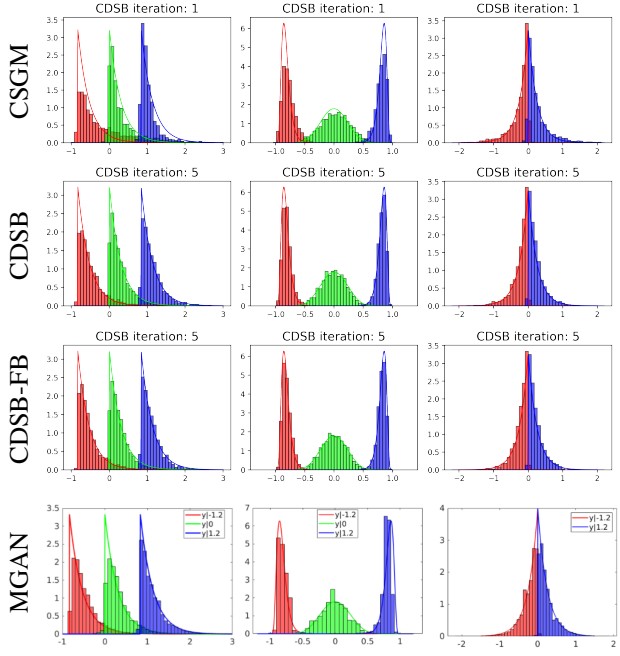

Figure 2: True posterior $p(x|y^{\text{obs}})$ for $y^{\text{obs}} \in \{-1.2, 0, 1.2\}$ (solid lines) and approximations for the 2D examples.

| | | MCMC | CDSB | CDSB-FB | CDSB-C | MGAN | IT |
|---|---|---|---|---|---|---|---|
| Mean | $x_1$ | .075 | .066 | .068 | **.072** | .048 | .034 |
| | $x_2$ | .875 | .897 | .897 | **.891** | .918 | .902 |
| Var | $x_1$ | .190 | .184 | **.190** | .188 | .177 | .206 |
| | $x_2$ | .397 | .387 | .391 | **.393** | .419 | .457 |
| Skew | $x_1$ | 1.94 | **1.90** | 2.01 | **1.90** | 1.83 | 1.63 |
| | $x_2$ | .681 | .591 | .628 | .596 | **.630** | .872 |
| Kurt | $x_1$ | 8.54 | 7.85 | **8.54** | 8.00 | 7.64 | 7.57 |
| | $x_2$ | 3.44 | 3.33 | **3.51** | 3.27 | 3.19 | 3.88 |

Table 1: Estimated posterior moments for the BOD example. The closest estimates to MCMC are highlighted in bold.

We run CDSB on each of the examples with 50,000 training points and compare with the Monotone GAN (MGAN) algorithm [Kovachki et al., 2021]. CDSB uses a neural network model with 32k parameters (approximately 6x less parameters than MGAN) with $N = 50$ diffusion steps. Figure 2 shows the resulting histogram of the learned $p(x|y^{\text{obs}})$ and the true posterior for $y^{\text{obs}} \in \{-1.2, 0, 1.2\}$. As can be observed, the empirical density of CDSB samples is sharper and aligns more closely with the ground truth density. We also observe that using more CDSB iterations corrects the sampling bias compared to using only one CDSB iteration (which corresponds to CSGM). Using forward-backward sampling (CDSB-FB) further improves the sample quality.

## 7.2 BIOCHEMICAL OXYGEN DEMAND MODEL

We now consider a Bayesian inference problem on biochemical oxygen demand (BOD) from Marzouk et al. [2016].

Let $X_1, X_2 \overset{\text{i.i.d.}}{\sim} \mathcal{N}(0, 1)$, $A = 0.8 + 0.4 \operatorname{erf}(X_1/\sqrt{2})$, $B = 0.16 + 0.15 \operatorname{erf}(X_2/\sqrt{2})$ and $Y = \{Y(t)\}_{t=1}^5$ satisfy $Y(t) = A(1 - \exp(-Bt)) + Z$ with $Z \sim \mathcal{N}(0, 10^{-3})$. Table 1 displays moment statistics of the estimated posterior $p(x|y)$ (standard deviations are reported in the supplementary), in comparison with the "ground truth" statistics computed using $6 \times 10^6$ MCMC steps as reported in Marzouk et al. [2016]. To match the evaluation in Kovachki et al. [2021], the reported statistics are computed using 30,000 samples and averaged across the last 10 CDSB iterations. The resulting posterior displays high skewness and high kurtosis, but all CDSB-based methods achieve more accurate posterior estimation than MGAN and the inverse transport (IT) method in Marzouk et al. [2016].

## 7.3 IMAGE EXPERIMENTS

### 7.3.1 Gaussian Reference Measure

We now apply CDSB to a range of inverse problems on image datasets. We consider the following tasks: (a) MNIST 4x SR (7x7 to 28x28), (b) MNIST center 14x14 inpainting, (c) CelebA 4x SR (16x16 to 64x64) with Gaussian noise of $\sigma_y = 0.1$, (d) CelebA center 32x32 inpainting. For CSGM-C and CDSB-C, we consider the following choices for conditional $p_{\text{ref}}(x|y)$: for tasks (a) and (c), we use the upsampled $y$ directly as described in Section 5.1; for inpainting tasks (b) and (d), we use a separate neural network with the same architecture as $\mathbf{F}, \mathbf{B}$ to output the initialization mean. In Table 2 we report PSNR and SSIM (the higher the better), as well as FID scores (the lower the better) for RGB images only. We display a visual comparison between the methods in Figures 3 and 4, and additional image samples in the supplementary. CDSB and CDSB-C both provide significant improvement in terms of quantitative metrics as well as visual evaluations, and high-quality images can be generated quickly under few iterations $N$.

### 7.3.2 Pre-trained SR Model for Reference Measure

We further explore here the possibility of using a non-Gaussian $p_{\text{ref}}(x|y)$ to further bridge the gap towards the true posterior $p(x|y)$. We utilize the super-resolution model SRFlow [Lugmayr et al., 2020], which produces a probability distribution over possible SR images using a conditional normalizing flow. We use their pre-trained model checkpoints for the 8x SR task for CelebA (160x160). We then train a short CDSB model with SRFlow as $p_{\text{ref}}(x|y)$, in order to take advantage of the high sampling quality of diffusion models. As can be seen from Figure 5, with only $N = 10$ steps the CDSB model is able to make meaningful improvements to the SRFlow samples, especially in the finer details such as facial features and hair texture. Quantitatively, CDSB-C produces significant improvement over

|  | $N=5$ | $N=10$ | $N=10$ | $N=20$ | $N=20$ | $N=50$ | $N=20$ | $N=50$ |
|---|---|---|---|---|---|---|---|---|
| CSGM | 17.22/0.672 | 20.03/0.795 | 14.77/0.599 | 16.31/0.706 | 19.52/0.471/92.02 | 20.52/0.567/48.68 | 24.22/0.844/17.62 | 25.29/0.878/7.18 |
| CDSB | 18.55/0.746 | 20.69/0.792 | 16.24/0.618 | 16.61/0.657 | 19.72/0.504/57.22 | 20.70/0.590/40.08 | 24.88/0.850/19.85 | 26.61/0.894/3.87 |
| CSGM-C | 18.61/0.749 | 20.83/0.838 | 16.38/**0.701** | 16.53/0.730 | 20.44/0.566/44.44 | 20.84/0.592/22.89 | **28.26**/0.914/3.63 | **28.14**/0.913/1.31 |
| CDSB-C | **19.67**/**0.753** | **20.95**/**0.840** | **16.60**/0.700 | **16.65**/**0.747** | **21.11**/**0.614**/28.41 | **21.46**/**0.646**/13.71 | 28.19/**0.915**/2.28 | 28.06/**0.914**/1.14 |
| | (a) | | (b) | | (c) | | (d) | |

Table 2: Results for (a) MNIST 4x SR; (b) MNIST 14x14 inpainting; (c) CelebA 4x SR with Gaussian noise; (d) CelebA 32x32 inpainting. Reported results are denoted in the format PSNR↑/SSIM↑(/FID↓).

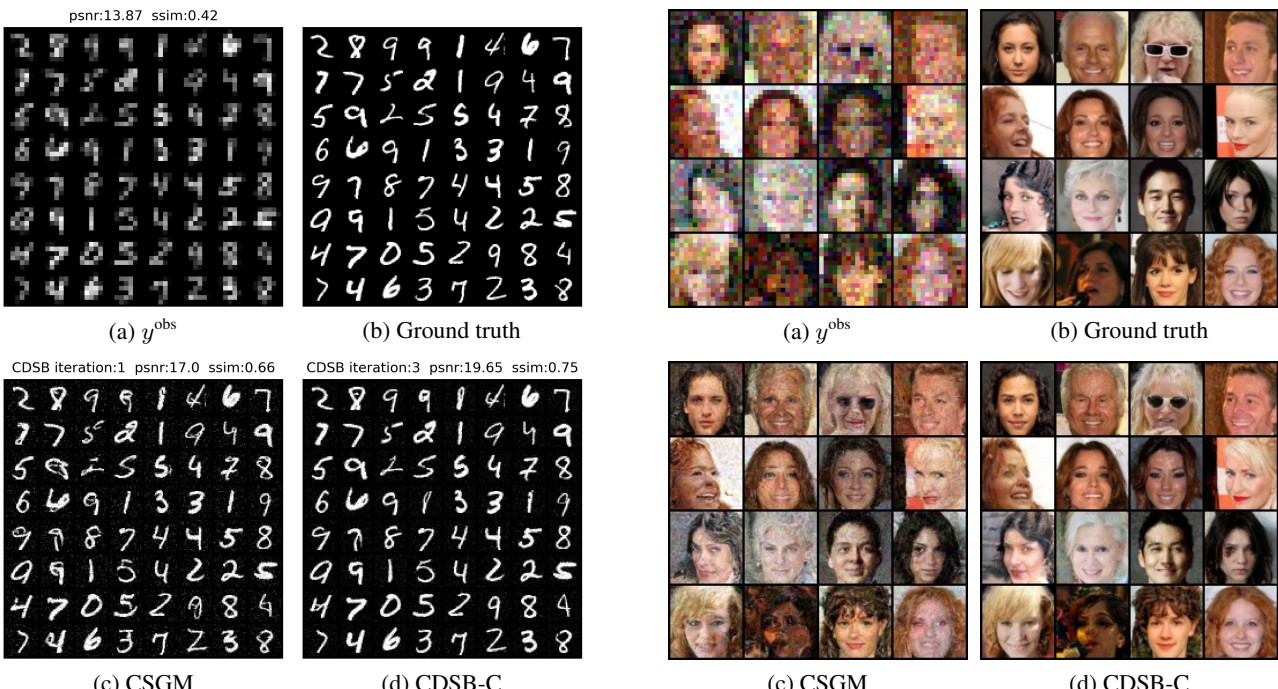

psnr:13.87  ssim:0.42

(a) $y^{\text{obs}}$     (b) Ground truth

CDSB iteration:1  psnr:17.0  ssim:0.66    CDSB iteration:3  psnr:19.65  ssim:0.75

(c) CSGM     (d) CDSB-C

Figure 3: Uncurated samples for the MNIST 4x SR task with $N=5$.

(a) $y^{\text{obs}}$     (b) Ground truth

(c) CSGM     (d) CDSB-C

Figure 4: Uncurated samples for the CelebA 4x SR with Gaussian noise task with $N=20$.

the FID score at the cost of a decrease in PSNR; see Table 3. Note that this choice of non-Gaussian $p_{\text{ref}}(x|y)$ is not compatible with CSGM. Interestingly CSGM-C still improves the PSNR compared to SRFlow, but produces worse FID scores than CDSB-C and blurry samples.

### 7.4 FILTERING IN STATE-SPACE MODELS

Consider a state-space model defined by a bivariate Markov chain $(X_t, Y_t)_{t\geq 1}$ of initial density $\mu(x_1)g(y_1|x_1)$ and transition density $f(x_{t+1}|x_t)g(y_{t+1}|x_{t+1})$ where $X_t$ is latent while $Y_t$ is observed. We are interested in estimating sequentially in time the filtering distribution $p(x_t|y_{1:t}^{\text{obs}})$, that is the posterior of $X_t$ given the observations $Y_{1:t} = y_{1:t}^{\text{obs}}$. We show here how CDSB can be used at each time $t$ to obtain a sample approximation of these filtering distributions. This

CDSB-based algorithm only requires us being able to sample from the transition density $f(x_{t+1}|x_t)g(y_{t+1}|x_{t+1})$ and is thus more generally applicable than standard techniques such as particle filters [Doucet and Johansen, 2009].

Assume at time $t$, one has a collection of samples $\{X_t^i\}_{i=1}^M$ distributed (approximately) according to $p(x_t|y_{1:t}^{\text{obs}})$. We sample $X_{t+1}^i \sim f(x_{t+1}|X_t^i)$ and $Y_{t+1}^i \sim g(y_{t+1}|X_{t+1}^i)$. The resulting samples $\{X_{t+1}^i, Y_{t+1}^i\}_{i=1}^M$ are thus distributed according to $p_{\text{join}}(x_{t+1}, y_{t+1}) := p(x_{t+1}, y_{t+1}|y_{1:t}^{\text{obs}})$. We can also easily obtain samples from $p_{\text{jref}}(x_{t+1}, y_{t+1}) := p_{\text{ref}}(x_{t+1}|y_{t+1}, y_{1:t}^{\text{obs}})p(y_{t+1}|y_{1:t}^{\text{obs}})$ where $p_{\text{ref}}(x_{t+1}|y_{t+1}, y_{1:t}^{\text{obs}})$ is an easy-to-sample distribution designed by the user. Thus we can use CDSB to obtain a (stochastic) transport map between $p_{\text{join}}(x_{t+1}, y_{t+1})$ and $p_{\text{jref}}(x_{t+1}, y_{t+1})$ and applying it to $Y_{t+1} = y_{t+1}^{\text{obs}}$, we can obtain new samples from $p(x_{t+1}|y_{1:t+1}^{\text{obs}})$. A similar strat-

| $p_{\text{ref}}(x|y)$ | CSGM-C | CDSB-C |
|---|---|---|
| Gaussian | 22.21/0.521/87.02 | 23.86/0.628/31.65 |
| SRFlow $\tau = 0.8$ | **24.97**/0.701/26.83 | 24.34/0.674/**15.00** |
| SRFlow $\tau = 0.8$ | | 24.83/**0.702**/30.92 |

Table 3: Results for CelebA 8x SR. Reported results are denoted in the format PSNR↑/SSIM↑/FID↓. The final row reports our evaluated results of the SRFlow model.

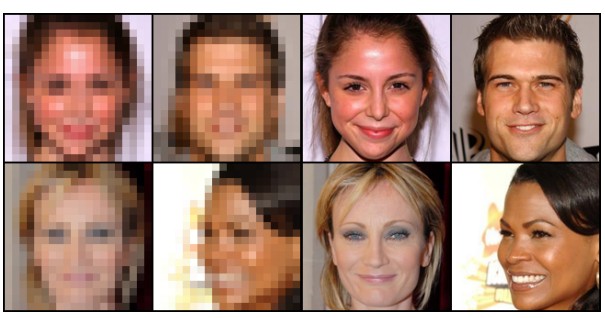

(a) $y^{\text{obs}}$        (b) Ground truth

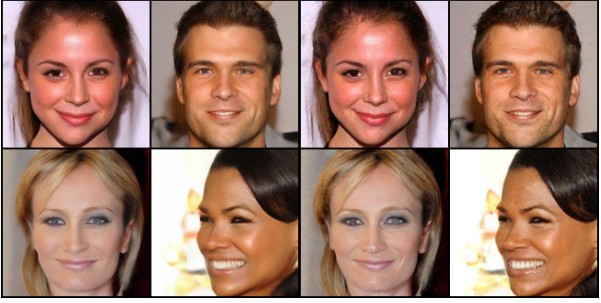

(c) SRFlow        (d) CDSB-C

Figure 5: Paired samples for CelebA 8x SR. The SRFlow samples (c) are inputted as conditional initialization into CDSB-C (d), which produces fine modifications over $N = 10$ steps (Best viewed when zoomed in).

egy for filtering based on deterministic transport maps was recently proposed by Spantini et al. [2022].

We apply CSGM and CDSB to the Lorenz-63 model [Law et al., 2015] following the procedure above for a time series of length 2000. We consider a short diffusion process with $N = 20$ steps, as well as a long one with $N = 100$. To accelerate the sequential inference process, in this example we use analytic basis regression instead of neural networks for all methods, and we only run 5 iterations of CDSB. As the EnKF is applicable to this model, we can use the resulting approximate Gaussian filtering distribution it outputs for $p_{\text{ref}}(x_{t+1}|y_{t+1}, y^{\text{obs}}_{1:t})$ in CSGM-C and CDSB-C.

Table 4 shows that for $N = 20$ both CDSB and CDSB-C successfully perform filtering and outperform the EnKF, whereas both CSGM and CSGM-C fail to track the state accurately and diverge after a few hundred times steps.

CDSB-C achieves the lowest error consistently. When using $N = 100$, CSGM can achieve RMSE comparable with CDSB-C using $N = 20$, but CDSB still provides advantages compared to CSGM. CSGM-C achieves comparable RMSE as CDSB-C with suitably long diffusion process in this case. For lower ensemble size, e.g. $M = 200$, occasional large errors occur for some of the runs; see supplementary for details. We conjecture that this is due to overfitting.

| $M$ | 500 | 1000 | 2000 |
|---|---|---|---|
| EnKF | .354±0.006 | .355±.005 | .354±.003 |
| CSGM(-C) (short) | | Diverges | |
| CDSB (short) | .251±.011 | .218±.008 | .196±.005 |
| CDSB-C (short) | **.236±.012** | **.207±.014** | **.178±.007** |
| CSGM (long) | .232±.008 | .203±.009 | .182±.009 |
| CDSB (long) | .220±.012 | .195±.007 | .166±.004 |
| CSGM-C (long) | **.210±.009** | **.185±.005** | .162±.004 |
| CDSB-C (long) | .218±.014 | **.185±.008** | **.160±.003** |

Table 4: RMSEs over 10 runs between each algorithm's filtering means and the ground truth filtering means for $N = 20$ (short) and $N = 100$ (long).

# 8 DISCUSSION

We have proposed a SB formulation of conditional simulation and an algorithm, CDSB, to approximate its solution. The first iteration of CDSB coincides with CSGM while subsequent ones can be thought of as refining it. This theoretically grounded approach is complementary to the many other techniques that have been recently proposed to accelerate SGMs and could be used in conjunction with them. However, it also suffers from limitations. As CDSB approximates numerically the diffusion processes output by IPF, the minimum $N$ one can pick to obtain reliable approximations is related to the steepness of the drift of these iterates which is practically unknown. Additionally CSGM and CDSB are only using $y^{\text{obs}}$ when we want to sample from $p(x|y^{\text{obs}})$ but not at the training stage. Hence if $y^{\text{obs}}$ is not an observation "typical" under $p_{\text{obs}}(y)$, the approximation of the posterior can be unreliable. In the ABC context, the best available methods rely on procedures which sample synthetic observations in the neighbourhood of $y^{\text{obs}}$. It would be interesting but challenging to extend such ideas to CSGM and CDSB. Other interesting potential extensions include developing an amortized version of CDSB for filtering that would avoid having to solve a SB problem at each time step, and a conditional version of the multimarginal SB problem.

### Acknowledgements

We thank James Thornton for his helpful comments. We are also grateful to the authors of [Kovachki et al., 2021] for sharing their code with us.

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
