# OpenReview forum: "Conditional Simulation Using Diffusion Schrödinger Bridges"
_auai.org/UAI/2022/Conference — UAI 2022 Poster_

### Official Review · Reviewer_Zmsd · 2022-04-12

**Q2(1) Originality/Novelty:** 2
**Q2(2) Significance/Impact:** 2
**Q2(3) Correctness/Technical Quality:** 3
**Q2(6) Clarity Of Writing:** 3
**Q6 Overall Score:** 6
**Q8 Confidence In Your Score:** 4

**Q1 Summary And Contributions:**

The paper proposes conditional diffusion schodinger bridges, which extends regular schodinger bridges to modeling p(x | y), through pairs of (x, y) from the dataset and/or some simulation process. The paper proposes an "expected" DSB problem as the conditional DSB problem. This also allows the use of conditional reference measure (CDSB-cond) that may accelerate sampling. Empirical results on datasets such as MNIST, CelebA illustrate the effectiveness of the method.

**Q2 Assessment Of The Paper:**

More detailed information regarding each of these aspects is given below:

**Q2(4) Quality Of Experiments (Optional):**

2: Fair: The experimental evaluation is weak: important baselines are missing, or the results do not adequately support the main claims.

**Q2(5) Reproducibility:**

2: Fair: Key resources (e.g., proofs, code, data) are unavailable but key details (e.g., proof sketches, experimental setup) are sufficiently well-described for an expert to confidently reproduce the main results.

**Q3 Main Strengths:**

1. The paper aim to argue why the original DSB formulation cannot be directly used here.

2. Empirical results show that CDSB outperforms some CSGM method, as well as other baselines on 4 tasks.

**Q4 Main Weakness:**

1. Given the existence of many works on conditional diffusion models, conditional SGM, the technical novelty of CDSB seems to be less amazing -- the solution seems to be quite straightforward, just applied to the slightly different DSB case.

2 The empirical results, especially on images, are quite lacking.

3. The paper can illustrate more about the necessity of using DSB in cases where it could be better than diffusion models, such as learning transport maps between arbitrary distributions.

4. Clarity of the proposed method can be improved.

5. [Not a weakness, but it could be better to clarify] Why is the proposed method much more efficient than previous SGM or DSB-based methods?

**Q5 Detailed Comments To The Authors:**

Adding to point 1:
 - In fact, the "inability" in the DSB method may be viewed as arbitrary, since one can always choose a ref distribution that can be sampled from (just choose p(x0) to be Gaussian independent from y)?

Adding to point 2:
 - Again, there are many baselines for tasks such as image super-resolution that are not discussed at all; even in the context of diffusion models, there is SR3, which I believe may achieve better results than what we see here.
 - SNIPS [Kawar et al., 2021] is also a competitive baseline using SGMs and the only difference is that they have a known forward model for low-res images. While it is helpful to have cases where mapping from x to y is not known (and has to be learned), I think the comparison should be raised to the main paper instead of the appendix.
 - It would be nicer if a case is shown where forward mapping has to be learned.

Adding to point 3:
- The advantage of the DSB formulation is the ability to learn from p_ref that are non-Gaussian, which also has a "optimal" transport interpretation. This is shown in the paper by De Bortoli. It would be nice to add more experiments on this front since that would be a more compelling reason to use DSB instead of conditional diffusion models.

Adding to point 4:
- While it is true that the paper has a lot of background to cover, I believe that some details should be better emphasized in order to be relatively self-contained. For example, despite the argument that CDSB requires a lot less iterations in many cases, I have failed to see a detailed description of the sampling algorithm or how the noise levels are chosen for a target iteration number (which is in the back corner in the appendix). I think you should use the space to discuss this more and list the sampling algorithm in the main paper, instead of listing algorithm 1, which is almost the same as algorithm 2 (save for the additional y part). Unless the reader has deep knowledge regarding all the details of the previous related work, the main paper would probably leave a lot of questions in their mind.

**Q7 Justification For Your Score:**

+ The exact same method has not been proposed before.
+ The method seems to be quite efficient compared to earlier baselines on SGM and DSB.

- Weak experiments, especially in images where there are many baselines that should be compared against.
- Experiments do not fully motivate the use of DSB over SGM / diffusion-based methods.
- Exposition is sometime unclear in the main text, the key ideas are simple but drowned in a sea of math.


**Q9 Complying With Reviewing Instructions:**

1: Yes.

---

### Official Review · Reviewer_uBPG · 2022-04-13

**Q2(1) Originality/Novelty:** 3
**Q2(2) Significance/Impact:** 3
**Q2(3) Correctness/Technical Quality:** 3
**Q2(6) Clarity Of Writing:** 2
**Q6 Overall Score:** 6
**Q8 Confidence In Your Score:** 3

**Q1 Summary And Contributions:**

Denoising diffusion models are successful generative models. Unconditional and conditional simulations have both been studied extensively, and it is known that they are computationally intensive in the generative process.
The Schrödinger bridge method has previously been proposed to reduce the computational cost during generation for unconditional simulation. The article adapts the Schrödinger bridge approach to conditional simulation in the formulation of denoising diffusion models.

**Q2 Assessment Of The Paper:**

More detailed information regarding each of these aspects is given below:

**Q2(4) Quality Of Experiments (Optional):**

3: Good: The experimental evaluation is adequate, and the results convincingly support the main claims.

**Q2(5) Reproducibility:**

2: Fair: Key resources (e.g., proofs, code, data) are unavailable but key details (e.g., proof sketches, experimental setup) are sufficiently well-described for an expert to confidently reproduce the main results.

**Q3 Main Strengths:**

The article presents some degree of novelty in that it adapts the Schrödinger bridge approach to conditional simulation in the formulation of denoising diffusion models. This approach has been previously studied in the unconditional case, and it is of interest to provide a conditional counterpart. This is not a priori a trivial task. Conditional simulation in the framework of denoising diffusion models is a problem of relevance, and it is known to be computationally intensive. The approach suggested here, which adapts a solution to the issue of computational cost in the unconditional case to the conditional one, is therefore of potential relevance. The experiments seem to support the aims of the study.

**Q4 Main Weakness:**

It seems that the main contribution is to replace the direct adaptation of the Schrödinger bridge approach given in equation (5) with an averaged and more manageable formulation, which is given in equation (6).
The exposition could be improved by including more definitions and more explanations on the intuitive reasoning, deferring a more detailed account to the supplementary material. The supplementary material provides several further perspectives (this is to some extent a strength as well), but they seem to be expressed a bit vaguely.

**Q5 Detailed Comments To The Authors:**

The method developed in this article should be probably presented as a separate section. It seems that section 3 includes both known results, and the results that are introduced in this article. Although contextualizing the contributions is generally a good practice, the main results of the article should go separated. The formulation of SB  for conditional simulation could be exposed as an unanswered problem in section 3, and a formulation of the solution provided in this article could be given in the next section for instance.
Also, it seems that the whole idea behind SB is not really explained. A clear mention to optimal transport and its role in SB does not seem to be included. For instance, in the reference by De Bortoli et al., the beginning of the article consists of a careful overview of the main concepts, including an intuitive explanation. Something like this is missing here. See for instance figure 1 in the same reference. It really helps to clarify things. The main body of the paper would do better with a more intuitive/descriptive approach, where equations are referred to in the supplementary materials. Of course this can be carried out only to some extent, but trying would probably improve the layout.
The results in the supplementary material use several specific results from other sources. The latter are well specified (very detailed references), but it might be helpful for the reader to also include a description of the result that is being used. Something like “recall that from [....] it follows that”. The reader needs to jump between references excessively. This does not help understanding. Providing a brief explanation of the result that is being used allows the reader to understand the overall picture, complete the relevant part of the proof, and later double check the details of the reference. The first paragraph of section A seems to be incomplete.

**Q7 Justification For Your Score:**

As previously stated, the problem addressed is of relevance, and the authors have provided enough details regarding the implementation as well as theoretical considerations, however the results seem to be somewhat similar in spirit to other references.

**Q9 Complying With Reviewing Instructions:**

1: Yes.

---

### Official Review · Reviewer_t9Mm · 2022-04-15

**Q2(1) Originality/Novelty:** 3
**Q2(2) Significance/Impact:** 3
**Q2(3) Correctness/Technical Quality:** 3
**Q2(6) Clarity Of Writing:** 3
**Q6 Overall Score:** 7
**Q8 Confidence In Your Score:** 2

**Q1 Summary And Contributions:**

This paper extends the Diffusion Schroedinger bridge approach, proposed in the context of (unconditional) score based generative modeling [1] to the conditional setting to accelerate the diffusion step.  In addition, improvements are proposed arising from the conditional formulation (e.g. from the conditional initialization rather than starting from noise). Results are shown for toy, 2D synthetic examples and with MNIST and CelebA for superresolution and inpainting.


**Q2 Assessment Of The Paper:**

More detailed information regarding each of these aspects is given below:

**Q2(4) Quality Of Experiments (Optional):**

3: Good: The experimental evaluation is adequate, and the results convincingly support the main claims.

**Q2(5) Reproducibility:**

2: Fair: Key resources (e.g., proofs, code, data) are unavailable but key details (e.g., proof sketches, experimental setup) are sufficiently well-described for an expert to confidently reproduce the main results.

**Q3 Main Strengths:**

- Generally, the work is quite well written and the pieces are well put together.
- The paper does a good job going over the context of the work, that diffusion probability models take a large number of steps to converge to the reference distribution in the forward step and acceleration efforts are therefore beneficial.
- The experiments are sensibly put together, with toy synthetic examples, and results for inpainting and superresolution. It is demonstrated that the use of the conditional diffusion Schroedinger bridge can get comparable results as the conditional score based generative model, but at much smaller iteration counts.

**Q4 Main Weakness:**

I would have appreciated some more material on conveying the intuitions behind the original (DSB) method and its extension to the conditional setting - for instance, connection with optimal transport, in continuous/discrete setting analogues.

**Q5 Detailed Comments To The Authors:**

- Could the authors clarify in more detail the forward-backward sampling procedure and the intuitions behind it?
- How do the inpainting results look for CelebA (they are shown for MNIST).

**Q7 Justification For Your Score:**

The paper goes about extending the unconditional Schroedinger Bridge algorithm to the conditional equivalent in a principled way. Experiments shown back up the idea, showing equivalent quality of generations over much fewer iterations using the Schroedinger Bridge as compared with the vanilla conditional Score based model.

**Q9 Complying With Reviewing Instructions:**

1: Yes.

---

### Decision · Program_Chairs · 2022-05-15

**Decision:**

Accept (Poster)

**Comment:**

Meta Review: Pros:
Original approach to generalize the Schroedinger bridge formulation to the conditional setting, accelerating training of high-fidelity conditional generative models at lower computational cost.
Cons:
Comparison to baselines is limited. Better intuition about the method could be provided.

Quality: Good
Clarity: Good
Originality: Fair-Good
Significance: Fair-Good